# Using drivers and transmission pathways to identify SARS-like coronavirus spillover risk hotspots

Renata L. Muylaert [1] ✉, David A. Wilkinson[2], Tigga Kingston [3], Paolo D'Odorico [4], Maria Cristina Rulli [5], Nikolas Galli [5], Reju Sam John[6], Phillip Alviola[7] & David T. S. Hayman [1]

The emergence of SARS-like coronaviruses is a multi-stage process from wildlife reservoirs to people. Here we characterize multiple drivers—landscape change, host distribution, and human exposure—associated with the risk of spillover of zoonotic SARS-like coronaviruses to help inform surveillance and mitigation activities. We consider direct and indirect transmission pathways by modeling four scenarios with livestock and mammalian wildlife as potential and known reservoirs before examining how access to healthcare varies within clusters and scenarios. We found 19 clusters with differing risk factor contributions within a single country (N = 9) or transboundary (N = 10). High-risk areas were mainly closer (11-20%) rather than far (< 1%) from healthcare. Areas far from healthcare reveal healthcare access inequalities, especially Scenario 3, which includes wild mammals and not livestock as secondary hosts. China (N = 2) and Indonesia (N = 1) had clusters with the highest risk. Our findings can help stakeholders in land use planning, integrating healthcare implementation and One Health actions.

The process of infectious disease emergence from animals begins with the cross-species transmission (spillover) of a microbe (e.g. virus, bacterium, fungus, protozoa) to a new animal host in which it is pathogenic[1–3]. Identifying areas of higher spillover probability is an important strategy for pandemic prevention and has largely focussed on estimating host distributions[4,5] and modeling frameworks for adding proxies for disease risk and spread in the face of limited data[4–6]. Yet, successful emergence events are complex multi-stage processes with many possible pathways leading from the original wildlife reservoir to sustained transmission in people[7]. The probability of any of these pathways occurring and resulting in infection emergence varies temporally and spatially, so cross-scale mapping of the multiple, diverse drivers of disease emergence is needed to better allow decision-makers to know where to focus surveillance and mitigation strategies[8].

Human infectious diseases almost all came from other species[3]. COVID-19, Ebola virus disease, Mpox, HIV/AIDS, and Zika virus disease are recent examples, whereas those like measles arose after the Neolithic Agricultural Revolution[9]. Zoonotic disease emergence has accelerated in recent decades, likely as a result of diverse interacting drivers such as accelerated land use change[10], human encroachment of natural habitats, increasing and changing contacts among and between wildlife and domestic animals, and has been mostly linked to mammals and birds[11]. Bats are among the natural hosts of viruses in the coronavirus (family *Coronaviridae*) subgenus *Sarbecovirus* (severe acute respiratory syndrome (SARS)-related coronaviruses), which

[1] School of Veterinary Science, Massey University, Palmerston North, New Zealand. [2]UMR ASTRE, CIRAD, INRAE, Université de Montpellier, Plateforme Technologique CYROI, Sainte-Clotilde, La Réunion, France. [3]Department of Biological Sciences, Texas Tech University, Lubbock, TX, USA. [4]Department of Environmental Science, Policy, and Management, University of California, Berkeley, Berkeley, CA, USA. [5]Department of Civil and Environmental Engineering, Politecnico di Milano, Milan, Italy. [6]Department of Physics, Faculty of Science, University of Auckland, Auckland, New Zealand. [7]Institute of Biological Sciences, University of the Philippines- Los Banos, Laguna, Philippines. ✉e-mail: R.deLaraMuylaert@massey.ac.nz

includes SARS-CoV-1 and SARS-CoV-2, the cause of SARS and COVID-19[12,13]. Bat hosts of sarbecoviruses are broadly distributed but the highest diversity is in Southeast Asia[5]. Human infection with *Sarbecovirus* from bats may be more frequent than reported from traditional surveillance[14] and likely includes secondary hosts[15,16]. Viral infection prevalence contributes to the risk of spillover[2] and can be influenced by biological factors such as birthing cycles[17,18] and external stimuli such as human changes to land use[19], though these influences may vary by host, virus, or location[20,21].

The One Health approach acknowledges the interconnection of human, animal, and environmental health, aiming to address health challenges and prevent them holistically[22]. Large-scale risk assessments in which areas with similar risk profiles are identified provide invaluable information to inform One Health actions[12,23] and can be rapid, while the development of local, detailed, and intricate spillover and outbreak risk assessments are costly and can take a long time[24,25]. Since detailed and validated data for recent reports on outbreak risk reduction are lacking for most regions of the globe (e.g. the Sendai framework[26]), a broad evaluation targeting *Sarbecovirus* emergence can be advantageous to discuss diverse contexts across the region where most natural hosts of sarbecoviruses occur. Human encroachment has led to decreased distances between bat roosts and human settlements[27], so part of the relevant hazard for inferring spillover risk can be spatially quantified from remotely sensed proxies for socio-ecological risk factors.

Here, we identify where putative drivers for emergence risk overlap, focusing on the biological possibility of the emergence of a *Sarbecovirus*. Our goal is to aid mitigation and surveillance activities throughout South, East, and Southeast Asia through a One Health approach by identifying both where efforts should focus and which risk factors should be prioritized based on where outstanding values overlap the most (hotspots of spillover potential) and based on healthcare access. Specifically, we asked: Where are the hotspots of spillover risk that capture a range of hypothesized transmission scenarios representing different pathways for the emergence of a novel SARS-like coronavirus? Can we identify spatially cohesive clusters of hypothesized risk drivers that, when combined, increase the risk of zoonotic spillover[23,28]? How does access to healthcare for high-risk areas vary according to transmission scenarios?

We predicted that there would be co-occurring hotspots for most risk drivers converging in biodiverse regions with bat hosts in regions with the greatest pressure from anthropogenic land use change, regardless of country boundaries. The four scenarios evaluated represent different nested transmission pathways for sarbecoviruses in bats to infect people. We assume that the risk of emerging new SARS-like outbreaks is associated with social, biological, and environmental components, and because there are unobserved dynamics for emerging viruses[29], we evaluated four nested spillover pathway scenarios based on landscape change and potential hosts[30]. Scenario 1 represents a direct transmission from known bat Sarbecovirus hosts to people. This transmission would be facilitated by the landscape condition, human population, and known bat hosts. Although investigations suggest that direct transmission of sarbecoviruses from bats to humans may be possible[31], it has yet to be better documented[14,32,33]. Rather, the involvement of an intermediary or bridging host appears more likely[34], perhaps because this allows for recombination and viral evolution and/or leads to greater exposure to human populations. Consequently, we developed Scenarios 2–4 to represent indirect pathways that build on Scenario 1 (Fig. 1). In Scenario 2, we consider indirect transmission by adding livestock as intermediate hosts, and in Scenario 3, we consider indirect transmission by adding wild mammals as intermediate hosts. Finally, in Scenario 4, we consider indirect transmission, including both these indirect pathways, so this final scenario comprises landscape conditions, human population, known bat hosts, mammalian livestock, and wild mammals (Fig. 1). We

expected clusters to occur across the country borders and differing pathways to alter risk hotspot distributions.

## Results

### Characterization of risk driver hotspots

The study region comprises a 25796-pixel grid for the terrestrial area evaluated. Univariate hotspot areas differ in magnitude (Fig. 1) and extent. Most hotspots concentrate at latitudes between 20° and 40°. The univariate hotspots with the largest spatial extent are those obtained for agricultural and harvest land, followed by high-integrity forests and areas with high deforestation potential. Notably, hotspot patterns were insensitive to an increase in the critical threshold value for defining a hotspot, showing marked patterns both at 95% and the 99% percentiles (Supplementary Fig. 1), though coldspots decreased (Supplementary Fig. 2). The majority of the included region comprises coldspots for primary bat hosts. Drivers with the greatest extent of coldspots were livestock (pigs and cattle), followed by known bat hosts. The largest extent of intermediate areas was for built-up land, which presented no cold spots due to the ubiquitous nature of human occupation in terrestrial areas. The largest differences in results for all Bovidae livestock versus cattle-only hotspots (see Methods) are in central China, parts of the north (Hebei, Shanxi, and Henan) China, and central India (Supplementary Fig. 3). The complete overlap of hotspots considering all ten univariate hotspots at one grid cell never occurred.

### Scenarios

Regardless of the scenario, the largest hotspot overlaps occur in central and southeast China, south and northwestern India, and Java. Differences between Scenario 1 with direct transmission from known *Sarbecovirus* bat hosts to people and Scenario 4 with indirect transmission via livestock and wild mammals are largest in central China (Fig. 2). The largest differences between each scenario and Scenario 1 (the scenario with the fewest covariates) concentrated in central and southern China. Scenario 3, with wildlife but no livestock intermediate hosts, had the least differences in relation to Scenario 1. Similar to Scenario 1, Scenario 2 shows most hotspot convergences in central and south China. For Scenario 4 (indirect transmission−all mammals), the most important PCA axes show a clear 'natural axis' and an anthropogenic axis, where the pig production layer is intermediate to the influence of both axes (Supplementary Fig. 4). Both main axes explain 58.7% of the total variation (PC1 = 33.5%, PC2 = 24.8%). Maximum overlap for non-human potential primary and secondary hosts occurred across China and Vietnam. While strong commonalities remain, the differences between scenarios highlight how the involvement or not of domestic or other wildlife hosts alters the risk profile.

### Hotspot co-occurrence in clusters

We identified spatially cohesive clusters, including all hypothesized risk drivers (Fig. 3). We used a multivariate hierarchical partitioning algorithm to infer clusters of similar values in the region. To find the optimal number of clusters, we inspected the total within-cluster sum of squares variation from iterations of up to 40 clusters, in addition to inspecting the optimal number of clusters given by the max-p algorithm. The optimal number of multivariate spatial clusters is nine when 10% of the human population is used as a minimum bound variable and 19 for 5% of the human population. There is an incremental benefit reduction in iterations with more clusters from nineteen groups (Supplementary Fig. 5). Moreover, because the clusters from the cut-off value of 5% are nested within the 10% clusters (Supplementary Fig. 6), we discuss the 19 clusters in the main text. The detected clusters were commonly transboundary (Table 1), located across a maximum of six countries and a median of 2 countries. Besides 10 transboundary clusters, nine clusters are restricted to a single country: 6 in China, 2 in India, and 1 in Indonesia (Java).

 

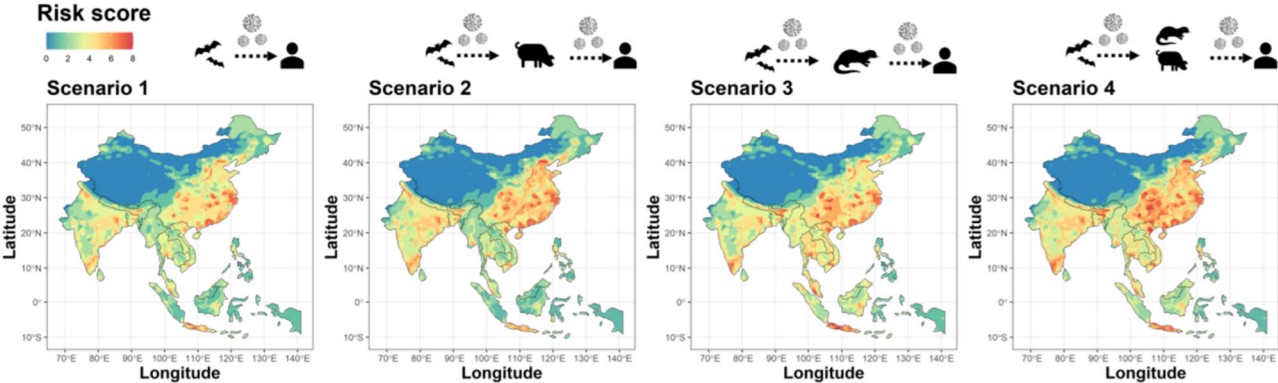

**Fig. 1 | Hotspots of potential factors contributing to the emergence of SARS-like coronaviruses. A** Spatial distribution of hotspots based on putative drivers of risk of *Sarbecovirus* emergence was evaluated in four scenarios. **B** List of variables per scenario marked as black dots and proportion (%) of areas classified as hotspots, intermediate or coldspots across the study region, including wildlife, landscape change, livestock, and exposure in humans. This classification used a critical threshold value at the 0.95 percentile to define hotspots and coldspots.

**Fig. 2 | Emergent risk scores for scenarios containing co-occurring drivers associated with landscape change and zoonotic pathogen emergence.** Landscape, human population, and known bat *Sarbecovirus* hosts are included in all models and are the sole drivers in Scenario 1, representing direct bat-to-human transmission. To incorporate indirect transmission through secondary hosts, mammalian livestock are included in Scenario 2, wild mammals in Scenario 3, and both mammalian livestock and wild mammals in Scenario 4. The internal white area in China represents no data values for Lake Qinghai.

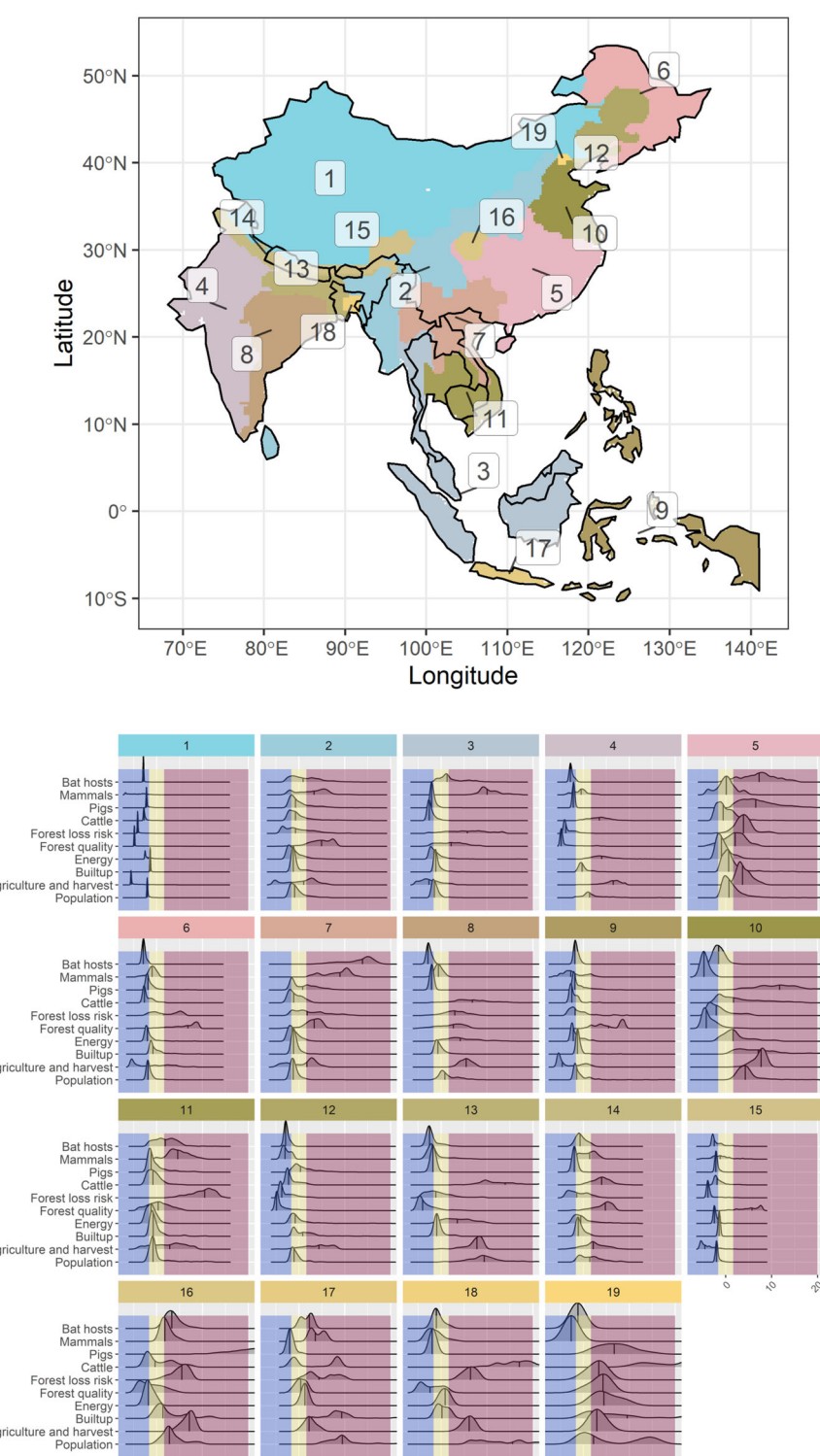

**Fig. 3 | Distribution of clusters of risk factors associated with potentially new emerging SARS-like coronaviruses. The values include all potential mammalian hosts, land use change, and human exposure density distributions** (Scenario 4). Areas located in the red zone represent hotspots, yellow zones are intermediate areas, and coldspots are in blue at a 95% alpha error level. A map with country (black lines) labels is shown in Supplementary Fig. 8.

**Table 1 | Identification of spatial clusters and the number of variables for which the median were coldspots, intermediate, or hotspots (n = 190)**

| Cluster ID | Countries | Number of countries | Coldspot | Intermediate | Hotspot |
|---|---|---|---|---|---|
| 1 | Bhutan, China, India, Nepal | 4 | **9** | 1 | 0 |
| 2 | Bangladesh, Bhutan, China, India, Sri Lanka, Myanmar | 6 | 0 | **8** | 2 |
| 3 | Brunei, Indonesia, Lao PDR, Myanmar, Malaysia, Thailand | 6 | 3 | 4 | 3 |
| 4 | India | 1 | 4 | 3 | 3 |
| 5 | China | 1 | 0 | 5 | 5 |
| 6 | China | 1 | 6 | 3 | 1 |
| 7 | China, Cambodia, Lao PDR, Myanmar, Thailand, Vietnam | 6 | 0 | 5 | 5 |
| 8 | India | 1 | 2 | 3 | 5 |
| 9 | Philippines, Indonesia, East Timor | 3 | **7** | 2 | 1 |
| 10 | China | 1 | 5 | 0 | 5 |
| 11 | Cambodia, Lao PDR, Philippines, Thailand, Vietnam | 5 | 0 | **6** | 4 |
| 12 | China | 1 | 5 | 4 | 1 |
| 13 | Bangladesh, Bhutan, India, Nepal | 4 | 4 | 2 | 4 |
| 14 | Bhutan, China, India, Nepal | 4 | 1 | **6** | 3 |
| 15 | Bhutan, China, India, Myanmar, Nepal | 5 | **8** | 1 | 1 |
| 16 | China | 1 | 2 | 2 | **6** |
| 17 | Indonesia | 1 | 1 | 2 | **7** |
| 18 | Bangladesh, India | 2 | 2 | 4 | 4 |
| 19 | China | 1 | 1 | 1 | **8** |
|  | | Total | 60 | 62 | 68 |

The top three values for each hotspot, intermediate, and coldspot are in boldface.

The top-ranking clusters in terms of hotspots were all located in a single country (China or Indonesia): Beijing (cluster 19), Java (cluster 17), and Sichuan and Yuzhong District, Chongqing (cluster 16). The clusters with the highest scores were among the smaller clusters in geographical extent. Inner-West China (cluster 1), South Lhasa and Arunachal Pradesh (cluster 15), and Philippines, East Timor, and West Papua (cluster 9) had the highest scores for coldspots. Areas with the highest scores for the Intermediate class were Assam, West Burma block, Steppe, and Sri Lanka (cluster 2), followed by Southwest Indochina (cluster 11) and North India (cluster 14). Clusters with the all Bovidae livestock version are in Supplementary Fig. 7, and they were very similar to the cattle-only versions from the main text, except for the Beijing area and the division of the two larger clusters in India, West India, and East India.

**Potential outbreak detection and spread**

When we match the risk factor spatial information with healthcare access measured as travel time[35], the largest differences between combinations of quantiles of the two covariates are in the lowest and highest quantiles of both variables (Fig. 4a, Supplementary Movie 1). We calculated the areas with high-risk values that are far or close to healthcare for all scenarios (Supplementary Fig. 9) within the spatial clusters from the skater analysis. From the entire study region, areas closer to healthcare that had high hotspot overlap (areas in yellow in Fig. 4) covered an area ranging from 11.96% in Scenario 1 to 20.28% in Scenario 2, 14.66% in Scenario 3 and 13.67% in Scenario 4. Areas far from healthcare that have high hotspot overlap (in red Fig. 4) were much rarer and varied according to scenarios, always covering less than 1% of the studied region, ranging from 0.1% in Scenario 1 to 0.30 in Scenario 2, 0.91% in Scenario 3—with significantly higher travel times to healthcare (Fig. 4b, $p < 0.05$)—and 0.22% in Scenario 4 (Supplementary Table 1, Fig. 4b). The relationship between travel time to healthcare and human population counts (Supplementary Fig. 10) shows that areas far from healthcare tend to have lower population counts (our proxy for exposure), but the relationship is non-linear.

## Discussion

Urgent actions are needed to decrease disease emergence risk[36,37]. Using a macroscale approach, we assessed the distribution of locations with a greater risk of experiencing *Sarbecovirus* spillover events using landscape conditions and exposure of potential hosts (wildlife, domestic, human). Landscape conditions coupled with predictions of the distribution of known hosts and proxies for potential hosts and processes linked to human exposure to novel viruses can be a powerful tool for sample prioritization when limited viral spillover information is available, such as for sarbecoviruses[14].

The overlap of risk factor hotspots represents pressure points on natural ecosystems that have been extensively altered in terms of agriculture, deforestation, and livestock production. In some cases, these clusters still have high values for forest quality and known bat *Sarbecovirus* host diversity (for instance, cluster 5—central China, and cluster 17—Java, Indonesia). Areas, where outstanding values of different risk factors converge can pose a severe risk to disease emergence and biological conservation. In Sichuan, China—cluster 16— values of livestock production are extremely high, and largely extensive farming takes place concomitantly with the presence of hotspots for mammal diversity (including higher values for known bat hosts) and very high deforestation risk. Unfortunately, deforestation rates and increasing demand for livestock are evident in our top-rated clusters[4], within biodiversity-rich areas, with high forest loss risk and a very large human population (e.g. in the cases of Beijing—cluster 19 and Java—cluster 17).

The similarity of hotspots at 95% and 99% percentiles suggests that our analysis was robust to uncertainties in the definition of hotspots. However, cold spots significantly decreased at 99%, losing space to intermediate areas. In terms of influence on risk scores, since we focus on hotspots, the increase in intermediate areas did not influence our risk metric. However, it is important to add this sort of sensitivity analysis, especially in regional studies applying this type of assessment or when prioritization is clearly dependent on hotspot conditions. We assume that intermediate areas in proximity to hotspots and where socio-ecological transitions, such as those related to the livestock

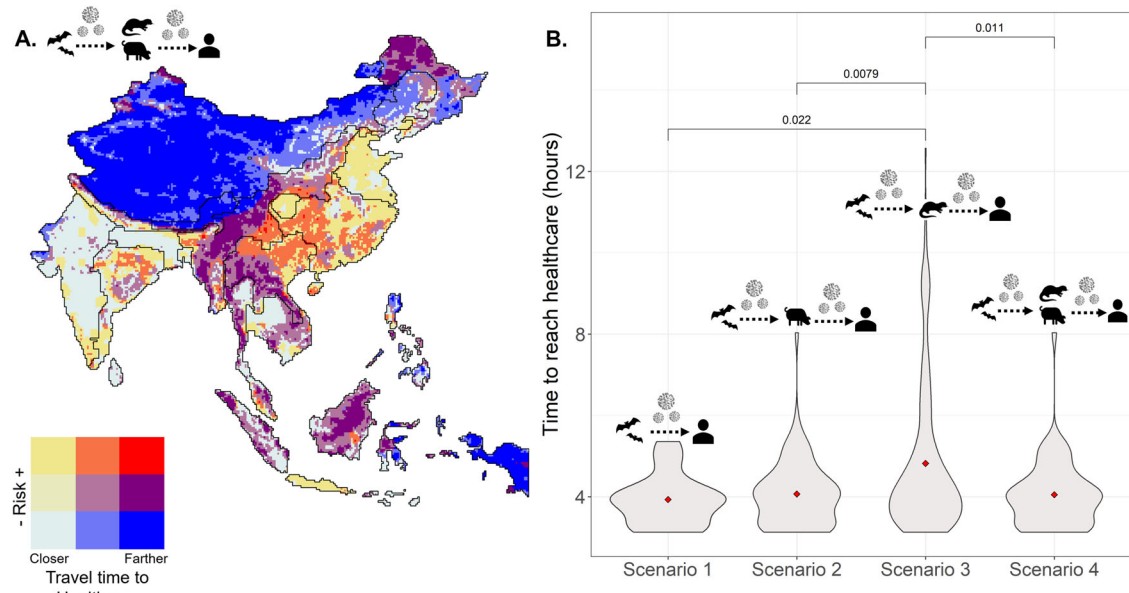

**Fig. 4 | Bivariate map showing the risk scores from hotspot data and access to healthcare. A.** Black lines divide the limits for the 19 clusters identified by the multivariate spatial cluster analysis; Scenario 4 is represented in the map. **B.** Time to reach healthcare in areas where high emergent risk co-occurred far from healthcare for each of the four scenarios. Average values are in red, with p-values showing which pairwise comparisons differ from the null expectation of no difference in Wilcoxon's test. Landscape, human population, and known bat Sarbecovirus hosts are included in all models and are the sole drivers in Scenario 1, representing direct bat-to-human transmission. To incorporate indirect transmission through secondary hosts, mammalian livestock are included in Scenario 2, wild mammals in Scenario 3, and both mammalian livestock and wild mammals in Scenario 4.

revolution, are at the greatest risk of transitioning to hotspots[4]. Even without a complete transition to hotspots, clusters with mostly intermediate values for stressors have had zoonotic spillovers of coronaviruses in the past[15,32,33], notably those in central China cluster 2 and on their edges with cluster 7 (north Lao PDR, north Vietnam, south China). Further, there is an overlap of several identified clusters with areas that concentrate hosts of other viruses with pandemic potentials, such as the Nipah virus[38]. The intermediate and high-risk areas within clusters need a multidimensional approach to mitigation that combines targeted surveillance of human populations, and other animals and environment monitoring, taking a One Health approach[39]. In addition to surveillance and biosecurity (discussed below), these approaches emphasize nature-based mitigation strategies, looking at the socio-economic drivers that shape local landscape conditions. Our analyses also show that risk factor clusters are commonly multinational. Action plans are likely complex tasks to implement, therefore, transboundary, coordinated action between nations that share territorial limits is paramount if the configuration of hotspots is taken into account when managing, protecting, and restoring land to mitigate disease emergence risk.

Remote areas with greater travel times to healthcare that have little spatial overlap with risk factor hotspots (blue, Fig. 4) may represent conditionally safer areas. In those areas, the priority should be assessing and reducing other disaster and disease risks. In areas of high potential assessed risk (khaki, orange, and red, Fig. 4), actions should be focused on the drivers of spillover. Recent literature suggests three broad, cost-effective actions to minimize pandemic risk: better surveillance of pathogen spillover, better management of wildlife trade, and substantial reduction of deforestation (i.e., primary prevention)[37]. Landscape planning should have priority, as these can have other co-benefits[40,41] and can include preventive measures to reduce levels of contact between people and potential wild and domestic animal hosts. Biosecurity measures and surveillance and integrated wildlife monitoring[42] are also key where multi-component risk levels are higher[43]. Syndromic, virological, serological, and behavioral risk surveillance of people with regular proximity to known

reservoir or potential amplifier hosts[43] can be of great value in these hotspots, but the ultimate prevention should be in primary prevention[44]. Beyond viral monitoring and discovery, prevention can be achieved by reducing deforestation, managing livestock production and wildlife trade, and increasing sustainable management of agricultural areas[37].

Surveillance effort correlates with detecting infections, and where human populations intersect with wildlife, risk increases[45,46]. In addition, evidence from Brazil also suggests zoonotic risk increases with remoteness (along with increased wild mammal species richness) and decreases in areas with greater native forest cover[47]. Our results suggest high-risk areas are often (11–20%) associated with faster travel times to healthcare, compared to remote areas (<1%) (yellow and red respectively, Fig. 4). The problem posed by remote sites for emergence mitigation is that while spillover probability and initial ease of spread may be lower, so too is detection probability[45] because of the distance to healthcare. This may allow localized, remote outbreaks to establish and spread in human populations before detection[48–50]. Our findings can be helpful in allocating efforts for surveillance, sustainability, and conservation actions and long-term plans for ecological intervention, including in areas with high emergent risk scores. Importantly, additional layers of prioritization could be added to implement mitigation actions on hotspots, for instance, where climate change vulnerability is also high, such as in Java[51]. Also, regions of China are outstandingly connected due to high mobility[52,53], which highlights the need to reduce pressures arising from multiple hotspots.

Scenario 2 (indirect transmission through livestock) had the highest number of regions with high-risk areas close to healthcare (yellow, Fig. 4). These areas are extensive across the study region in all scenarios and should be prioritized for temporal screening for viruses in livestock, the understanding of known hosts, and investments in improving public health responses to spread. High-risk areas far from healthcare (red) represent small regions of our study area (<1%) in all scenarios, where Scenario 1 had the fewest and Scenario 3 had the highest areas. These are areas with higher possibilities for spillover that would also be likely to go undetected during the early stages of human-

to-human transmission and spread. In those regions, urgent action to prevent contact, reduce deforestation, and enhance biodiversity protection should take place, as well as improvements in healthcare access. Human populations that are more vulnerable to risks could be targets for equitable distribution of promising solutions, such as pan-coronavirus vaccines[54].

Our findings are a snapshot of macroscale spatial trends that can be used for prioritizing more detailed analysis depending on the context and policy priorities. The United Nations Development Program (UNDP) recommends the creation of 'Maps of Hope' for maintaining essential life support areas[55], but the relationship between biodiversity loss, fragmentation, and zoonotic disease is seldom considered in the designation of such areas. We advocate for a One Health approach in which the risks of pathogen emergence are explicitly integrated into initiatives addressing habitat management, restoration, and protection[55], and have demonstrated that this risk can be mapped at large scales with insights into variability in the distribution of key drivers[22].

We acknowledge the complexity of pathogen responses to land use modification[56] and the important limitations of our findings. The datasets used here are all static yet global and accessible. The static nature of the datasets is one limitation in our assessment, as risk might vary temporally due to changes not apparent in static data. Hotspots may change in response to changes in economic and agricultural policies at national and subnational levels, international agreements such as Agenda 2030, and climate change adaptation[57]. There are also several empirical data limitations. For instance, although the data sources for the health facilities are generally accurate[35], omission errors can occur. Hospitals and clinics may close over time and while new ones are built, so it is important to take these limitations into consideration in local contexts, especially when access to healthcare is not possible using motorized vehicles. Regarding host data, cryptic diversity in bats[58] and uneven sampling occur for sarbecoviruses and their bat hosts[5], creating uncertainty regarding host-pathogen interactions that is difficult to account for. Ecological analyses at finer spatial and temporal scales than those used here can elucidate cascading events that result in zoonotic spillover. For example, Hendra virus spillover from bats to horses in Australia seems to be driven by interactions between climatic change altering the flowering phenology of important nectar sources, exacerbating food shortages resulting from native habitat loss and degradation, and nutritional stress in bats that may increase Hendra virus shedding[19]. Native resource declines have concurrently promoted the urbanization of many bat populations, increasing the human–bat interface and potential for spillover events to horses, which can act as intermediary hosts or even potentially direct to humans[59]. Our analyses may capture the macroscale processes but not these local events.

Similarly, while knowing that the top-priority traded mammals[60] are correlated with total mammalian diversity, local analyses should evaluate how factors that cannot be easily mapped or tracked deviate from the large-scale trend, such as animal trade and hunting, which is currently not feasible using a macroscale approach. Our workflow can, however, be easily coupled with detailed local data for spillover 'barriers' and host characteristics to bring insights and customize action plans, such as data on reservoir density, pathogen prevalence, pathogen shedding, and data on spillover recipients, such as susceptibility and infection[61]. This is especially important when macroscale and subnational level risk assessments are neither complete nor validated for most nations[62].

To date, the role of domestic intermediate hosts for sarbecoviruses is unclear, with numerous species able to be infected by SARS-CoV-2[63]. Here we include cattle and all Bovidae livestock evaluations, leading to similar overall results for clusters but with some univariate hotspots less intense, especially in central India and south China, while making them more intense around Beijing, highlighting how

uncertainties around host susceptibility and potential pathways leads to uncertainty regarding risk. The emergence of a novel coronavirus and re-emergence of a known *Sarbecovirus* through spillback is also possible[63] and may change risk profiles. Other factors that play a large role in outbreak response, such as conflict[64] and other societal challenges associated with health and the environment, might also be considered in local contexts.

The use of remote sensing layers can bring insights for land use planning when considering complex processes such as disease emergence. This process may benefit not only the understanding of risks but also local actions informed by broad patterns[6]. Recent models suggest that the implementation of smaller-scale land-use planning strategies guided by macro-scale patterns may help to reduce the overall burden from emerging infectious diseases[65], while also taking into account biodiversity conservation. This could be evaluated from multiple perspectives, including in the context of other planetary boundaries and how zoonotic disease risk sits within it[66], considering we have already passed the 1° warmer planet threshold[67].

In conclusion, we found evidence that hotspots of multiple conditions that contribute to the risk of emerging *Sarbecovirus* are commonly transboundary, but areas scoring the highest values of risk occurred in China and Indonesia. We also identify that most high-risk areas are not far from healthcare for each transmission scenario, but in all scenarios, there are several that are far from healthcare, especially Scenario 3 which includes wild mammals as secondary intermediate hosts. This work contributes to strengthening evidence of spatial clusters of multiple risk factors for disease emergence. We use a reproducible workflow based on hotspot analysis from broad-scale data that is accessible through open software and maps for easy interpretation. This can enable agencies to discuss and engage in new land-use planning actions by including stakeholders (academia, government, local communities, and non-governmental organizations) under a One Health perspective. The need to reduce inequalities in access to healthcare[68] without promoting encroachment into natural areas is a challenge. Efforts should focus on comprehensive land use planning, including the placement of healthcare facilities and other infrastructure[69]. Biodiversity provides essential ecosystem services, so primary prevention of spillover can benefit sustainability at multiple scales, sustaining life on earth and human health. Our findings can help stakeholders when evaluating multiscale policies, land use planning, and considering integrating community health programs into universal healthcare implementation[70] at transboundary, national, or subnational levels.

## Methods
We use South, East, and Southeast Asia (including western New Guinea) as our study region, where most *Sarbecovirus* hosts are concentrated[5,14] and where many unknown sarbecoviruses are estimated to exist[28]. We define our study region as the terrestrial area of the following countries: Bangladesh, Bhutan, Brunei, Cambodia, China, India, Indonesia, Lao PDR, Malaysia, Myanmar, Nepal, Philippines, Singapore, Sri Lanka, Thailand, Timor-East, and Vietnam.

### Characterization of univariate risk indicator hotspots
We identified spatial clusters of components of risk. We assume our inferred risk arises not from individual factors having outstanding high values (hotspots), but instead, it arises when they are combined, facilitating conditions for viral spillover. In that sense, our inference of risk is an emergent property of the system (emergent risk). We adopted a broad-scale risk estimation framework (https://mcr2030.undrr.org/quick-risk-estimation-tool) focusing on the potential for sarbecoviruses to emerge. The ten broad risk factors were five landscape-level conditions and five biological layers, according to four scenarios (see data sources and further descriptions and justification of these in Supplementary Table 2). The analysis is naive about the influence of

individual drivers on the risk of spillover in the sense that all factors were weighted equally in our scenario evaluations. We selected the following ten factors for land use change and landscape conditions: Intensity of (1) built-up land, (2) mining and energy, (3) agricultural and harvest land, (4) forest quality, and (5) local forest loss risk. As a measure of human or animal exposure, we used wildlife, comprising (6) known bat hosts and (7) all other wild mammals), (8) livestock (pigs and cattle), and (9) human populations. We use (10) the average of known sarbecovirus bat hosts as our primary host layer[5,14,71,72]. In all scenarios we also include human population counts because this increases the risk of contact and therefore transmission[73]. We included built-up areas, energy and mining, and agriculture and crop harvest landscape changes in all scenarios because these have been linked to increased coronavirus shedding in bats putatively due to stress and human contact rates[28,74]. We similarly used forest quality and risk of cover loss because emerging infectious disease risk is elevated in forested tropical regions experiencing land-use changes and where wildlife biodiversity (mammal species richness) is high[4,7,75,76]. We then used different potential secondary (aka intermediate) hosts to understand different interactions and transmission pathways, where we included pigs (Scenario 2, Scenario 4) because while SARS-CoV-2 transmission is unlikely[77], coronaviruses cause a variety of diseases in pigs[78], including infections-related to bat coronavirus[34]. We also included cattle (Scenario 2, Scenario 4), as coronaviruses caused infection[79] and disease in cattle[80], and coinfection can lead to recombination events[81]. We have also included other bovid livestock (Scenario 2, Scenario 4) because there are several regions in Southeast Asia where carabao (*Bubalus bubalis*) and other Bovidae livestock are more common than cattle (*Bos taurus*), so we provide results for all Bovidae livestock instead of cattle-only in the Supplementary materials. We also included wild mammals as potential secondary (aka intermediate) hosts, minus the known bat hosts (Scenario 3, Scenario 4), because EID risk is elevated in forested tropical regions experiencing land-use changes and where wildlife biodiversity (mammal species richness) is high. SARS-CoV-2 has also been detected in wildlife (spillback events)[63,75].

To avoid collinearity, we only selected variables with product-moment correlation coefficient (*r*) values < ±0.7 (Supplementary Fig. 11). The study region was divided into a spatial grid composed of 0.25 decimal degrees-sized tiles (~27 km). All variables were resampled to match this resolution. For data layers that were counted from shapefiles (other mammal species numbers), we applied median values for resampling. We ran a univariate hotspot analysis based on Getis-ord *G*$i$ values[82] considering each factor individually at 95% and 99% cut-off for critical values using rgeoda 0.0.9[83]. First, we created a list considering all data and then the 5 * 5 neighboring cells around these for the closest neighborhood (*n* = 25). Then, we ran a local *G* analysis on every pixel, assuming a two-sided alternative hypothesis at the 95% and 99% percentiles to check hotspot sensitivity to critical values. High-positive values indicate hotspot regions, and low negative values indicate coldspots. Pixels located in-between the alternative hotspot or coldspot hypothesis values are referred to as intermediate regions, where the value may reflect a random spatial process, i.e., no spatial clustering detected.

## Scenarios

Detected hotspots for all landscape condition components were used in combination with biological components in the scenario analyses. Scenario 1 considers direct transmission from bats to humans, where the biological risk is composed of the average number of bat species in which sarbecoviruses have been reported as the known primary hosts. For Scenario 2, we then considered the components of Scenario 1 in combination with potential intermediate hosts using: pig counts, cattle-only, or all Bovidae livestock counts. Scenario 3 considered bat hosts and the number of other wild mammal species present. For

Scenario 3, we used the wild mammal layer (minus known bat hosts) and known bat hosts as the potential intermediate (wild mammals) and source (known bat) hosts. We considered using a traded mammal layer instead of an all wild mammal layer in Scenario 3 because of evidence the first COVID-19 cases identified were linked to the Huanan Seafood Wholesale Market in Wuhan[16]. An available high-priority traded mammal layer, in which species were classified as traded by the International Union for the Conservation of Nature (IUCN)[60], however, is highly correlated (*r* = 0.864) with the wild mammal layer. Because of this correlation, in addition to high uncertainty regarding trade, we kept only the mammal layer and bat hosts layer in Scenario 3. A fourth scenario, Scenario 4, included all of the previous mammalian layers, be it wild or livestock, so combined all ten risk factors.

## Hotspot convergence in clusters

We evaluated the spatial clustering among hotspots, including all ten selected drivers (Scenario 4, five landscape descriptors, and five potential host components). We opted for doing a single cluster analysis because we cannot weigh the importance of the single variables for influencing an ultimate spillover event. The variables comprised here describe landscape conditions, human population, cattle, pig, bat hosts, and all other wild mammals. We assume areas that contain the most hotspots or that may be on the verge of becoming hotspots (intermediate areas) for the components evaluated are at higher risk of emerging new sarbecoviruses. A multivariate spatial cluster analysis was applied to Getis-ord *G*$i$ values[82] for every variable after the univariate hotspot analysis using rgeoda 0.0.9[83]. We used the multivariate skater (Spatial 'K'luster Analysis by Tree Edge Removal) hierarchical partitioning algorithm[84] to infer contiguous clusters of similar values in the region based on the optimal pruning of a minimum spanning tree. Spatial clusters represent emergent, cohesive risk combinations distributed in space. Contiguity was assessed by a queen weights matrix after transforming pixels to geographical coordinates. Distance functions were set to Euclidean. We evaluated the k number of clusters from 1 to 40. To find the optimal number of clusters, we evaluated the total within-cluster sum of squares variation, visually inspecting the point of inflection in the curve towards stabilization. As the reduction in increment was very smooth as clusters were added, we present the number of clusters for skater informed by the max-p algorithm. We used max-p to find the solution for the optimal number of spatially-defined clusters setting as a bounding variable (a variable that allows for a minimum value summed for each cluster) the human population amounts to 5% and 10%. The algorithm was computed at 99 interactions with 123456789 as a random seed.

To interpret the variation of hotspots between clusters, we counted the number of variables for which the median of the distribution is a hotspot (i.e., falling within the hotspot interval at 95% *Gi**). We then discuss the clusters based on the number of drivers that are already hotspots and the median values that fall in intermediate zones, so possibly closer to becoming hotspots and coldspots for each cluster. Finally, to understand the overall variation (and among clusters), we provide a principal component analysis (PCA) biplot through Scenario 4 to discuss major axes of variation between an optimal number of clusters. We ran the hotspot analyses with cattle-only and with the summed values for all Bovidae livestock (presented in the Supplemental Material). All geographical coordinates were warped to the World Mercator (EPSG: 3395) and World Geodetic System 1984 datum before spatial analysis.

## Emergent risk and its relationship with access to healthcare

After identifying the hotspots within the scenarios, we match their proximity to detection by matching the emergent risk score (i.e., number of hotspots) for every pixel with the level of motorized access to healthcare (hospitals and clinics)[35]. Access to healthcare measured as travel time was considered as both a proxy for connectivity and an

indicator of the likelihood of detection following infection spillover and spread. We use the word proximity to refer to the times when someone needs to reach a hospital or clinic: where far or distant areas mean it takes a longer time to reach them. We built bivariate maps and three-by-three quantile ($N = 9$) combinations considering the intensity of hotspots from their overlay and the values for access to healthcare, all rescaled from zero to one. We compared travel time in high-risk areas among scenarios using Wilcoxon's test. China has the largest number of pixels with healthcare facilities in the world, followed by other countries considered in our analysis, such as India, Indonesia, Thailand, and Malaysia. We assume that the travel time to healthcare dataset is adequately accurate for our study region since source data from Google Maps and Open Street Map data sources have robust quality controls[35]. More details on healthcare facility data coverage and accuracy in our study region are provided in Supplementary Tables 2 and 3. All analyses were done in QGIS 3.10.7[85], R 4.1.3[86], and bash[87].

## Data availability
The data processed in this study have been deposited in FigShare.

## Code availability
The code for the analyses can be found in GitHub and Zenodo.

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

## Acknowledgements

We thank Dr. Jinhong Luo and Marco A. R. Mello for their comments on earlier versions of the draft. Massey University's subscription to New Zealand eScience Infrastructure (NeSi) enabled us to use high-performance computing facilities. Te Pūnaha Matatini gave us support during the 2022 Mahia Te Mahi workshop. R.L.M., D.T.S.H., and R.S.J. are supported by Bryce Carmine and Anne Carmine (née Percival) through the Massey University Foundation; D.T.S.H. is supported by the Royal Society Te Apārangi, grant No. MAU1701.

## Author contributions

R.L.M. wrote the original draft. R.L.M. developed the code workflow. D.W. performed code revision. D.T.S.H. supervised the project. R.L.M., D.W., D.T.S.H, T.K., P.D., M.C.R., N.G., R.S.J., and P.A. contributed to developing the research questions, discussing the results, and paper revision.

## Competing interests

The authors declare no competing interests.
