## [Peer Review File · Nature Communications]

Using drivers and transmission pathways to identify SARS-like coronavirus spillover risk hotspotsREVIEWER COMMENTS

Reviewer #1 (Remarks to the Author):

In this article, Muylaert et al. develop a map of spillover risk for sarbecoviruses using a variety of drivers that all have clear ties to mechanisms involved in spillover. Their work relies on four scenarios of increasing biotic complexity.

I found the paper extremely well written and compelling, and I only have minor queries and comments about the accessibility.

In particular, it took me a while to unpack the last sentence of the introduction. I think this is because it never quite states the assumption that administrative borders are permeable to biological processes, and therefore may not be relevant when identifying the hotspots. I think this can be expanded by (i) clearly stating why the authors end up with this supposition and (ii) giving more breathing room to the role of the different pathways.

At l. 90, I think the introduction of hotspots might be nuanced a little, as the same word is used (not in this paper, but in this literature) to denote either hotspots of observed transmission, or hotspots of coevolutionary dynamics. I think this is a simple fix, and something like "hotspots of spillover potential" would be enough.

Getting back to the definition of hotspots (particularly in Fig. 1), it was unclear to me what the cutoff for a hotspot or a coldspot is. I think this is an important point as it can change our understanding of the risk, so this might be worth stating briefly how the pixels are assigned to cold/neutral/hot, and what the quantitative criteria is.

In a similar way, because this might change the result, I would encourage the authors to experiment with changing the value of these cutoffs, and report on how it affects the number/area of hotspots. This sort of sensitivity analysis would most likely reveal that the results are (qualitatively at least) robust to small uncertainties in the definition of cold/hot spots.

At line 105, it is not immediately clear what the 95% alpha error level refers to. I think this should be clarified in this sentence.

At line 131, I would similarly add a very short tidbit about the optimality criteria (which is absolutely the correct one, but it would save the readers a skip to the methods to verify it).

At line 133, I do not think this result is very surprising -- adding clusters decreases the cardinality of the clusters, and therefore starts increasing the residual variance. I think the formulation of "adding clusters" can make the process sound arbitrary, and this seems like this sentence would fare better in the methods.

At line 139 (I obviously enjoyed picking apart the clustering analysis), I was wondering whether this is surprising given the area of these countries? Is it more than we would expect if hotspot pixels were assigned at random? This would probably give readers a somewhat informative frame of reference for these values.

Reviewer #2 (Remarks to the Author):

This paper assesses environmental factors, or "drivers", associated with the spillover risk of Sarbecovirus (SARS-like coronaviruses) in South, South-east and East Asia, where SARS-like coronaviruses are prominent and unknown Sarbecovirus are estimated to exist. The authors identify and classify geographic hot spots or clusters associated with these risk factors, which is interesting and important.

Generally, this research would be interesting to readers in the field of infectious disease mapping and spatial epidemiology. However, several key issues should be improved before it can be considered for publication. I have outlined several major and minor suggestions below. If there is misunderstanding with the materials discussed in the paper, the authors are advised to improve clarity wherever necessary. Please also add line numbers in future drafts.

Introduction

1. "Viral infection prevalence contributes to the risk of spillover 2, and can be influenced by biological factors such as birthing cycles^{17,18} and external stimuli such as human changes to land use¹⁹ (but see^{20,21})." Please clarify what you mean by "but see" – is this supposed to highlight a counterargument? If so this should be stated in text rather than referenced.
2. The authors state, "Scenario 1 (direct transmission - known bat hosts) represents direct transmission from bats to people, facilitated by the landscape condition, human population, and known bat hosts. Although molecular investigations suggest that direct transmission of sarbecoviruses from bats to humans may be possible³⁰, it has yet to be better documented^{14,31,32}." If scenario 1 has not been documented, how likely is it for scenario 1 to happen in the near future? How is this weighted relative to the other scenarios?
3. I do not understand what constitutes the "global scenario" for Scenario 4. How is this different than the other scenarios?

Results

4. Fig 1 – it's not clear to me which panels refer to the four scenarios. I suggest the authors to clearly label which figures correspond to which scenarios. Can the authors consider removing groups (e.g., landscape change) from the x-axis and add them as coloured labels on the y-axis? To me this was less intuitive to follow.
5. Fig 3 – It would be useful if the authors add country labels to the map. I suggest the authors to move Fig 3 above Table 1 as it is more intuitive to contextualize the study regions visually. Panels need labelling.
6. What is the spatial resolution of access to travel time? This is not clear to me and impacts my interpretation of the bivariate map (map not "maps" in Fig 4 – please correct), and my understanding of how this is computed. In Methods, the authors state "After identifying the hotspots within the scenarios, we match their proximity to detection by matching the emergent risk score (i.e. number of hotspots) for every pixel with the level of motorized access to healthcare (hospitals and clinics)." Please define "proximity" and how this determined (e.g., threshold proximity).
7. What is the rationale for pairings for the Wilcoxon Test? Why are they compared to Scenario 3? This is not clear to me.

Discussion

8. The authors state, "The intermediate and high-risk areas within clusters need a multidimensional approach to mitigation that combines targeted surveillance of human populations, other animals and the environment with One Health approaches" . I appreciate that the authors have considered a One Health perspective but it would be helpful to provide some recommendations on what these "multidimensional" approaches would look like, especially when considering transboundary risk.
9. In the paragraph starting, "Remote areas that present little spatial overlap in risk factor hotspots (blue, Fig. 4) may represent conditionally safer areas...". Could the authors please offer some insight on how "reduction" and "prevention" can be achieved, for instance through existing case studies? Further, I appreciate that "reducing deforestation" is a suggestion, but there isn't really a discussion on how this can be achieved and what actions can be taken to support this.
10. In the paragraph starting, "We advocate for a One Health approach...". The One Health Approach isn't mentioned until the discussion. Could the authors please consider discussing the importance of this study for One Health in the Introduction?
11. In the section on limitations, references are needed on the Hendra virus spillover.
12. Could the authors please discuss limitations with the usage of remote sensing data, scale, and spatial biases for identifying disease hot spots and spillover? The robustness and limitations of the methods adopted are not clear to me.

Methods

13. The paper does not discuss or mention the data sources, data sets, and the metadata used for the analysis. This includes bats, cattle, other mammal, environmental factors, and healthcare access, to name a few. This is not transparent to the reader. Are the data, especially the land-use factors and primary and secondary host distributions, static or dynamic? Do the data contain geographic coordinates or exist as raster formats? It's unclear what preprocessing was conducted on the data prior to use. The authors adopted a 27km spatial grid and data were resampled to this scale. I am curious which data sets existed at lower and higher resolutions, and how aggregation (or disaggregation) are addressed to maintain the integrity of the original data.

14. Are the data sets open access? Can links be provided for the data sets in SI so the analysis can be reproduced?

15. It is unclear which software was used to conduct the Getis-ord G*I analysis and how spatial relationships are conceptualized. From the text it states "We created a list of closest neighbors considering all data and $n=25$ for the closest neighborhood." I understand this may imply nearest neighbour analysis but if so this needs to be specified and the analysis conducted to determine n should be explained. Was a sensitivity analysis conducted? If so, the authors should clarify.

16. According to the text, "We considered using a traded mammal layer instead of an all wild mammal layer in Scenario 3". I cannot find definitions of "traded mammal" and "wild mammal", and the overlap between these categories, in either the main text and SI. This would be important knowledge for wildlife conservation and surveillance purposes.

17. The authors selected 10 selected drivers for the analysis but I cannot find an explicit list of what they are in the Methods.

18. I do not understand the following sentence: "We used max-p to find the solution for the optimal number of spatially-defined clusters setting as a bounding variable (a variable that allows for a minimum value summed for each cluster) the human population amounts at 5% and 10%."

19. How is the travel time representing "access to healthcare" characterized? I wonder whether the most common mode of transportation ranges by region (e.g., car vs motobikes which are common in Southeast Asia), and how this is accounted for in the analysis. Is public transportation considered?

20. Are both public and private healthcare facilities considered? How many facilities are included? Is the proportion included representative of the countries? If not, I feel the authors should discuss how the completeness of the data adopted would impact their findings.

Reviewer #3 (Remarks to the Author):

Comments to Author(s)

This paper is generally interesting, sufficiently novel for publication, and makes a strong contribution to topical and important questions i.e. where are potential sars-like coronavirus spillover events most likely and how should the risk be characterised. It is well-written, although I would recommend checking carefully for grammatical errors and awkward phrasing in places.

The study uses secondary data analyses and the justification for the different data given in the supplementary material is considered and thorough. I would however like to see some more discussion in the main text of the potential biases and limitations of the data. For example, in the discussions regarding proximity to healthcare, is it likely that there may be more data for less remote areas, and that there may be biases in which areas have greatest uncertainty in terms of data and their proximity to major cities/healthcare facilities?

The analytical approach is well-explained and sound. My only query would be whether increased resolution would be possible for the maps presented, perhaps with some single-country maps e.g. for China and Indonesia. Also at least one map with the countries labelled individually would aid linking the maps to the text/tables for those less familiar with the region.

Some sensitivity analyses of the impact of the assumptions would be useful (perhaps in supplementary material), for example, it is assumed that intermediate areas are at risk of

becoming hotspots, how would results differ if this assumption were altered?

Another suggestion which may improve the context of the work, would be a few case studies of known emergence/spillover events and how the locations where these occurred would have been characterised within the frameworks described (this is mentioned briefly, but is a very interesting aspect that would help ground this work in real-world risk and enhance justification for taking this approach in more detail for most at-risk locations).

Overall I would recommend this paper for publication with minor revisions.

RESPONSE TO REVIEWER COMMENTS

Reviewer #1 (Remarks to the Author):

In this article, Muylaert et al. develop a map of spillover risk for sarbecoviruses using a variety of drivers that all have clear ties to mechanisms involved in spillover. Their work relies on four scenarios of increasing biotic complexity.

I found the paper extremely well written and compelling, and I only have minor queries and comments about the accessibility.

In particular, it took me a while to unpack the last sentence of the introduction. I think this is because it never quite states the assumption that administrative borders are permeable to biological processes, and therefore may not be relevant when identifying the hotspots. I think this can be expanded by (i) clearly stating why the authors end up with this supposition and (ii) giving more breathing room to the role of the different pathways.

R: Thank you, we appreciate your comments. We edited the introduction following your suggestion by explicitly stating the hotspots are regardless of national boundaries. We revised the final sentence for clarity. We also include a revised Figure 1 that helps readers visualise the scenarios..

At l. 90, I think the introduction of hotspots might be nuanced a little, as the same word is used (not in this paper, but in this literature) to denote either hotspots of observed transmission, or hotspots of coevolutionary dynamics. I think this is a simple fix, and something like "hotspots of spillover potential" would be enough.

R: Thank you. We revised this text as suggested.

Getting back to the definition of hotspots (particularly in Fig. 1), it was unclear to me what the cutoff for a hotspot or a coldspot is. I think this is an important point as it can change our understanding of the risk, so this might be worth stating briefly how the pixels are assigned to cold/neutral/hot, and what the quantitative criteria is.

R: Thank you. We include a statement early in the results, defining a hotspot as percentiles and we revised the methods to clarify our definition. Briefly, the Getis-Ord G^* statistic is a well-established method for cluster analyses and approximates to a z-score. We present the 95% quantiles in the main text along with the 99% added in this revised version in the supplementary text (see below) to define hotspots and coldspots, with the middle of the distribution considered to have no values that are significantly high or low clustered together.

In a similar way, because this might change the result, I would encourage the authors to experiment with changing the value of these cutoffs, and report on how it affects the number/area of hotspots. This sort of sensitivity analysis would most likely reveal that the results are (qualitatively at least) robust to small uncertainties in the definition of cold/hot spots.

R: Thank you for this idea. We now provide the hotspots using the 99% cutoff for identifying hotspots as a way of evaluating the sensitivity of the univariate hotspots to a stricter cut-off

value. We have added this additional result in the supplements. Notably hotspots are insensitive to changing the threshold from 95% to 99%, whereas large areas of coldspots shrink, losing space to intermediate areas, see figures below:

95%

99%

We believe the 95% cut-off provides more useful information from a risk management perspective. Moreover, because the univariate data is already highly skewed, we believe being more inclusive with the cutoff of 95% for the hotspots and coldspots makes sense to explore the contiguous clusters at the macroscale level. For instance, when we opt for a 99% cut-off, several variables will in turn stop having coldspots, and we believe that the categorisation of data in cold, intermediate, and hotspots is useful in terms of risk assessment with multiple covariates.

In terms of the cluster analysis, we used the maxp algorithm as a method to infer the optimal number of clusters, and this analysis stays the same regardless of cutoffs values, as it is fed by the z-scores as an input and it is purely quantitative from continuous values. We added text to the methods and this text to the discussion:

The similarity of hotspots at 95% and 99% percentiles suggest that our analysis was robust to uncertainties in the definition of hotspots. However, coldspots significantly decreased at 99%, losing space to intermediate areas. In terms of influence on risk scores, since we focus on hotspots, the increase in intermediate areas did not influence our risk metric. However, it is

important to add this sort of sensitivity analysis, especially in regional studies applying this type of assessment, or when prioritization is clearly dependent on hotspot conditions.

A new plot of the sensitivity analysis is now included as Supplementary Fig. 2:

At line 105, it is not immediately clear what the 95% alpha error level refers to. I think this should be clarified in this sentence.

R: Alpha error values are percentiles on frequentist tests. Values larger than the critical threshold value at 95% are hotspots and values lower than the critical value at 95% are cold spots, and values in the middle are intermediate areas. We clarified the text where needed.

We use the table provided in the original Ord and Getis paper to define those critical values, available here and now cited in the text: <https://onlinelibrary.wiley.com/doi/10.1111/j.1538-4632.1995.tb00912.x>

At line 131, I would similarly add a very short tidbit about the optimality criteria (which is absolutely the correct one, but it would save the readers a skip to the methods to verify it).

R: Good point. We have added a statement about that as follows:

We used a multivariate hierarchical partitioning algorithm to infer clusters of similar values in the region. To find the optimal number of clusters, we inspected the total within-cluster sum of squares variation from iterations of up to 40 clusters, in addition to inspecting the optimal number of clusters given by the max-p algorithm.

At line 133, I do not think this result is very surprising -- adding clusters decreases the cardinality of the clusters, and therefore starts increasing the residual variance. I think the formulation of "adding clusters" can make the process sound arbitrary, and this seems like this sentence would fare better in the methods.

R: True, but these are results and therefore we prefer to keep this sentence in the results. However, we have edited this part, so it does not sound arbitrary as it was an iterative process seeking to find an optimal number of clusters as the result. We show that there is an incremental benefit reduction in iterations with more clusters, from nineteen groups on (Supplementary Fig. 5). Moreover, because the clusters from the cut-off value of 5% are nested within the 10% clusters (Supplementary Fig. 6), we discuss the clusters for 19 areas in the main text.

At line 139 (I obviously enjoyed picking apart the clustering analysis), I was wondering whether this is surprising given the area of these countries? Is it more than we would expect if hotspot pixels were assigned at random? This would probably give readers a somewhat informative frame of reference for these values.

R: Thank you for your comment. The result that they cluster is not surprising, indeed it was one of our predictions, but we believe our clusters are useful to explore risk scenarios and possibilities for preparedness in different regions that might be more often be looking at administrative regions without considering transboundary conditions.

To respond to the point about comparing our result to random assignment, we performed an additional study for this reviewer response, but have not included it in the paper.

Briefly, in the text we evaluated clusters by looking at the ratio of between-group sum of squares to total sum of squares, iterating up to 40 clusters. The optimal number of clusters (n=19) when we set the population size as 5% as a bounding variable are clusters nested within the clusters (n=9) when we set 10% population size. We present and discuss the 19 optimal clusters in the main text.

To address the reviewers' comment, we generated 10 variables for randomly distributed continuous variables (simulating them from normal distributions centred on zero). Then, we ran the same cluster analysis we did, expecting that at our observed 19 optimal number of clusters, the simulated data would also result in a mix between transboundary clusters and single-country clusters. This analysis took approximately 24 hours, and resulted in a linear pattern for the random data, and never reached an asymptote on the number of clusters iterated from 1 to 40, meaning that there is no optimal number of clusters reached in simulation from random data. This is very different from what happens with empirical data, for which we always can evaluate an elbow pattern approximated to an asymptote.

Our data in “*skater*” shows decreasing added contributions from 19 clusters on in the top figure below, whereas the same analysis of the random simulation in *skater* never reaches an asymptote:

And this is how the clusters look from the simulation with random data, with 1 large cluster (gray) and mini-clusters scattered across the region:

Reviewer #2 (Remarks to the Author):

This paper assesses environmental factors, or “drivers”, associated with the spillover risk of Sarbecovirus (SARS-like coronaviruses) in South, South-east and East Asia, where SARS-like coronaviruses are prominent and unknown Sarbecovirus are estimated to exist. The authors identify and classify geographic hot spots or clusters associated with these risk factors, which is interesting and important.

Generally, this research would be interesting to readers in the field of infectious disease mapping and spatial epidemiology. However, several key issues should be improved before it can be considered for publication. I have outlined several major and minor suggestions below. If there is misunderstanding with the materials discussed in the paper, the authors are advised to improve clarity wherever necessary. Please also add line numbers in future drafts.

R: Thank you, we appreciate your comments and review. We have added line numbers.

Introduction

1. “Viral infection prevalence contributes to the risk of spillover 2, and can be influenced by biological factors such as birthing cycles^{17,18} and external stimuli such as human changes to land use¹⁹ (but see^{20,21}).” Please clarify what you mean by “but see” – is this supposed to highlight a counterargument? If so this should be stated in text rather than referenced.

R: Thank you, we have reworded this sentence to clarify that these are studies that have counter arguments.

2. The authors state, “Scenario 1 (direct transmission - known bat hosts) represents direct transmission from bats to people, facilitated by the landscape condition, human population, and known bat hosts. Although molecular investigations suggest that direct transmission of sarbecoviruses from bats to humans may be possible³⁰, it has yet to be better documented^{14,31,32}.” If scenario 1 has not been documented, how likely is it for scenario 1 to happen in the near future? How is this weighted relative to the other scenarios?

3. I do not understand what constitutes the “global scenario” for Scenario 4. How is this different than the other scenarios?

R: Thanks for your comment. We have clarified the text regarding the scenarios throughout the manuscript. The global scenario, Scenario 4, has all 10 drivers together, and it is the reference data for the cluster analysis. Since we cannot ensure which scenario is more likely to happen, we use the Scenario 4 as reference for the cluster analysis and give the same weights for the individual scenarios. We removed the term ‘global scenario’ from the text and simply refer to it as Scenario 4 now.

Results

4. Fig 1 – it’s not clear to me which panels refer to the four scenarios. I suggest the authors to clearly label which figures correspond to which scenarios. Can the authors consider removing groups (e.g., landscape change) from the x-axis and add them as coloured labels on the y-axis? To me this was less intuitive to follow.

R: Thank you for your suggestion, it was very helpful. We hope the new version of Figure 1 is more intuitive to follow now. We removed the groups as suggested and we added a panel so the reader can see clearly which variable is in which scenario.

5. Fig 3 – It would be useful if the authors add country labels to the map. I suggest the authors to move Fig 3 above Table 1 as it is more intuitive to contextualize the study regions visually. Panels need labelling.

R: As suggested, we moved Figure 3 above table 1.

We discussed adding country labels to the map, however we avoided this for two main reasons. First, we believe they make the figure look too busy. Second, several of these countries have borders that are contested or in conflict. The table in the main text lists the countries where each cluster occurs, without needing to highlight disputed borders.

We manually edited the coordinates of country labels, considering countries with multiple territories across islands such as Malaysia and Indonesia, and include the version with country labels (see below) in the supplements:

Clusters

6. What is the spatial resolution of access to travel time? This is not clear to me and impacts my interpretation of the bivariate map (map not “maps” in Fig 4 – please correct), and my understanding of how this is computed. In Methods, the authors state “After identifying the hotspots within the scenarios, we match their proximity to detection by matching the emergent risk score (i.e. number of hotspots) for every pixel with the level of motorized access to healthcare (hospitals and clinics).” Please define “proximity” and how this determined (e.g., threshold proximity).

R: Thank you, we corrected the typo.

The travel time layer was built based on spatially explicit data of observed hospitals and clinics across the world through Open Street Map and Google Maps, available at 1 km (Weiss et al. 2020) and warped to our working resolution (0.25 dd) through spatial resampling using the

bilinear method for interpolation. As stated, our intent was to evaluate patterns across broad areas, and we did not think it was a good idea to disaggregate predictions made on a coarser resolution to a finer resolution. This would not only add repeated information in finer scale data, but it would also make computational time for processing the data more challenging.

Lastly, we use proximity to refer to time proximity when someone needs to reach a hospital or clinic: far areas means it takes longer time to reach them, so we edited the methods to make this clearer.

Weiss, D.J., Nelson, A., Vargas-Ruiz, C.A. et al. Global maps of travel time to healthcare facilities. *Nat Med* 26, 1835–1838 (2020). <https://doi.org/10.1038/s41591-020-1059-1>.

7. What is the rationale for pairings for the Wilcoxon Test? Why are they compared to Scenario 3? This is not clear to me.

R: The test shows which pairwise comparisons differ from the null expectation of no difference. Time travel is higher for scenario 3. The other pairwise comparisons are not significant and thus not shown on the figure. We have improved the caption, so this is clear.

Discussion

8. The authors state, “The intermediate and high-risk areas within clusters need a multidimensional approach to mitigation that combines targeted surveillance of human populations, other animals and the environment with One Health approaches” . I appreciate that the authors have considered a One Health perspective but it would be helpful to provide some recommendations on what these “multidimensional” approaches would look like, especially when considering transboundary risk.

R: Thank you for this comment. This is an important point, but not the focus on the paper. However, we now provide a brief discussion of what those multidimensional approaches might look like. Specifically we mention biosurveillance, biosecurity, and nature based solutions, along with primary prevention (e.g. including reducing encroachment, deforestation, and some wildlife trade) with some further references. It is a complex topic, but we hope this is sufficient given the expected word limits and purpose of our study.

9. In the paragraph starting, “Remote areas that present little spatial overlap in risk factor hotpots (blue, Fig. 4) may represent conditionally safer areas...”. Could the authors please offer some insight on how “reduction” and “prevention” can be achieved, for instance through existing case studies? Further, I appreciate that “reducing deforestation” is a suggestion, but there isn’t really a discussion on how this can be achieved and what actions can be taken to support this.

R: Thank you for your comment. For brevity, we did not develop those ideas further, but we agree we can offer more insights on how reduction and prevention can be achieved. Please see our response above.

10. In the paragraph starting, “We advocate for a One Health approach...”. The One Health

Approach isn't mentioned until the discussion. Could the authors please consider discussing the importance of this study for One Health in the Introduction?

R: Thank you, we have added a sentence emphasizing the importance of our study to One Health in the introduction and citing this document:

One Health High-Level Expert Panel (OHHLEP) et al. One Health: A new definition for a sustainable and healthy future. PLoS Pathog. 18, e1010537 (2022).

11. In the section on limitations, references are needed on the Hendra virus spillover.

R: Thank you, we have now added Eby et al. (2023).

12. Could the authors please discuss limitations with the usage of remote sensing data, scale, and spatial biases for identifying disease hot spots and spillover? The robustness and limitations of the methods adopted are not clear to me.

R: We have added some discussion on limitations in the discussion. We now added some information on the limitations of the healthcare access layer and static remotely sensed data limitations in general.

Methods

13. The paper does not discuss or mention the data sources, data sets, and the metadata used for the analysis. This includes bats, cattle, other mammal, environmental factors, and healthcare access, to name a few. This is not transparent to the reader. Are the data, especially the land-use factors and primary and secondary host distributions, static or dynamic? Do the data contain geographic coordinates or exist as raster formats? It's unclear what preprocessing was conducted on the data prior to use. The authors adopted a 27km spatial grid and data were resampled to this scale. I am curious which data sets existed at lower and higher resolutions, and how aggregation (or disaggregation) are addressed to maintain the integrity of the original data.

R: Thank you for your comment. All the sources and their justification are described fully in the Supplementary Material. All sources are open sources, and we have downloaded them as rasters. The code for analysis is in the provided github repository at the end of the methods and the data is provided in a data repository. The resolution of every original layer is stated in the supplementary. Regarding pre-processing, we were not permissive to upscale data, so we used the appropriate resampling of original resolutions to our 27 km spatial grid to avoid repeated measurements.

14. Are the data sets open access? Can links be provided for the data sets in SI so the analysis can be reproduced?

R: Thank you. Yes, as above, all data sources are open access. Yes, the analysis can be reproduced. Data and code (R, bash) will be provided through open repositories with specific DOIs prior to publication.

15. It is unclear which software was used to conduct the Getis-ord G*I analysis and how spatial relationships are conceptualized. From the text it states “We created a list of closest neighbors considering all data and $n=25$ for the closest neighborhood.” I understand this may imply nearest neighbour analysis but if so this needs to be specified and the analysis conducted to determine n should be explained. Was a sensitivity analysis conducted? If so, the authors should clarify.

R: Thank you, we used the cited rgeoda R package. We have edited the text to clarify the method. We appreciate the comment on sensitivity, and we now have added a sensitivity analysis in terms of cut-off values for hotspots (0.95 and 0.99) using keeping the neighbourhood rules for both values constant as in our response to other comments above.

Now, being more specific about our reasoning for setting k as 25: The term k defines how many neighbours will be checked to determine the classification of a specific query point. For example, if $k=1$, the instance will be assigned to the same class as its single nearest neighbour. Since our resolution is 0.25 dd, we believe that a neighbourhood focused on hotspots should take into consideration all the neighbours around a pixel, including neighbors of neighbors, so we added the value 25 as a $5 * 5$ window of neighbourhood around every grid (24 neighbours, plus central grid = 25). We did not want to smooth it over larger windows, as we wanted a local hotspot statistic, and that is why we do not think it makes sense to iterate k over different values as a sensitivity analysis, as $k=25$ it is the closest value to a standard queen rule of spatial neighbourhood and gives us a sensible descriptor for local hotspots.

16. According to the text, “We considered using a traded mammal layer instead of an all wild mammal layer in Scenario 3”. I cannot find definitions of “traded mammal” and “wild mammal”, and the overlap between these categories, in either the main text and SI. This would be important knowledge for wildlife conservation and surveillance purposes.

R: Thank you. The definition of traded is defined by the IUCN. We revised the text to include this statement. The information on traded animals is from another article (Cronin et al. 2022). We agree could be helpful for wildlife conservation and surveillance purposes. However, since it was correlated with our mammal layer, we removed it from the analysis. We revised the text to clarify this.

Cronin, M. R., de Wit, L. A. & Martínez-Estévez, L. Aligning conservation and public health goals to tackle unsustainable trade of mammals. *Conservation Science and Practice* **n/a**, e12818 (2022).

17. The authors selected 10 selected drivers for the analysis, but I cannot find an explicit list of what they are in the Methods.

R: We edited the Methods to make sure the 10 selected drivers are clear.

18. I do not understand the following sentence: “We used max-p to find the solution for the optimal number of spatially-defined clusters setting as a bounding variable (a variable that allows for a minimum value summed for each cluster) the human population amounts at 5% and 10%.”

R: Thank you.

Briefly, the numerical solution for the optimal number of clusters depends on a bounding variable to inform the algorithm with a rule of how to arrange these clusters. This means that if you use 5% population as a rule, the clusters will cover an area where at least 5% of the population is recorded. Since population is highly skewed, we opted for using 5% and 10% as cut-offs from the bounding variable. Later on, by looking at variation minimization between both, we decided to keep the one with 19 clusters as the optimal solution. Taking 10% of the population makes the regions too large to explore details in the variation among drivers. However, we present both results in the supplements, for 9 and 19 clusters.

We have left the text unchanged, but hope this explanation helps.

19. How is the travel time representing “access to healthcare” characterized? I wonder whether the most common mode of transportation ranges by region (e.g., car vs motobikes which are common in Southeast Asia), and how this is accounted for in the analysis. Is public transportation considered?

R: Thank you for your question. It is a calculation of how long it takes to get to a hospital or clinic using a motorized vehicle, regardless of it being a public or private mean of transportation. This is stated in the text and the reference provided earlier in the text when this is first mentioned.

20. Are both public and private healthcare facilities considered? How many facilities are included? Is the proportion included representative of the countries? If not, I feel the authors should discuss how the completeness of the data adopted would impact their findings.

R: Yes, both are considered.

Please see here for the full details <https://www.nature.com/articles/s41591-020-1059-1>

Reviewer #3 (Remarks to the Author):

Comments to Author(s)

This paper is generally interesting, sufficiently novel for publication, and makes a strong contribution to topical and important questions i.e. where are potential sars-like coronavirus spillover events most likely and how should the risk be characterised. It is well-written, although I would recommend checking carefully for grammatical errors and awkward phrasing in places.

R: Thank you, we appreciate your review. We have carefully checked for grammar and awkward phrasing.

The study uses secondary data analyses and the justification for the different data given in the supplementary material is considered and thorough. I would however like to see some

more discussion in the main text of the potential biases and limitations of the data. For example, in the discussions regarding proximity to healthcare, is it likely that there may be more data for less remote areas, and that there may be biases in which areas have greatest uncertainty in terms of data and their proximity to major cities/healthcare facilities?

R: Thank you. We have added this discussion, and this point was also raised by another reviewer. Because data for clinics and hospitals were extracted through remote sensing, we believe there would actually be more uncertainty regarding clinics in the cities, but we now provide a better description of the limitations of the travel time layer and the role it plays in our findings.

The analytical approach is well-explained and sound. My only query would be whether increased resolution would be possible for the maps presented, perhaps with some single-country maps e.g. for China and Indonesia. Also at least one map with the countries labelled individually would aid linking the maps to the text/tables for those less familiar with the region.

R: Thank you.

As mentioned in response to another reviewer, we discussed adding country labels to the map prior to submission. However, we believe it made the figure in the main text look too busy and some national boundaries are contested. We have now added country labels but include the figure in the supplementary material.

Regarding the country level maps, we wanted to focus on natural, transboundary variation of factors regardless of administrative borders. Nevertheless, we appreciate the comments and provide a table in the main text listing the countries where each cluster occurs. One issue regarding figure resolution is that all figures have at least 400 dpi. In regard to spatial resolution, we treated this raster to match the resolution of the bat host predictions, which is 0.25 dd, through spatial resampling using the bilinear method for interpolation. Our intent was to evaluate patterns across broad areas, and we did not think it was a good idea to disaggregate predictions made on a coarser resolution to a finer resolution. This would not only add repeated information in finer scale data, but also significantly increase computational power required for processing the data.

Some sensitivity analyses of the impact of the assumptions would be useful (perhaps in supplementary material), for example, it is assumed that intermediate areas are at risk of becoming hotspots, how would results differ if this assumption were altered?

R: Thank you, that is a good point. We now present 2 cut-off values for the hotspots (0.95 and 0.99) and briefly discuss the impact of this variation. We present the 99% cut off for the hotspots as a figure in the supplements. Importantly, this would not alter the clusters by any means, as they are based on continuous Getis Ord-g values.

Another suggestion which may improve the context of the work, would be a few case studies of known emergence/spillover events and how the locations where these occurred would have been characterised within the frameworks described (this is mentioned briefly, but is a very interesting aspect that would help ground this work in real-world risk and enhance justification for taking this approach in more detail for most at-risk locations).

R: Thank you. One problem for coronaviruses is that the number of recorded events is so limited and the focus of our study is bat-borne coronaviruses. We have extracted some values based on this limited information (see coordinates below). From the locations we have for earliest detection by serology or molecular detection of the earliest cases for SARS-CoV-1 (SARS) and SARS-CoV-2 (COVID-19) the risk varies when we consider the bivariate maps for Scenario 4. Probably most importantly, 6 of 7 were in 'high risk' areas including the putative first cases for SARS and SARS-CoV-2. These two (SARS-CoV-1 and -2) are also in highly connected areas, close to healthcare facilities. However, we do not wish to over interpret this finding because there is uncertainty regarding these points, along with those of the other findings (especially serology, when infection may have occurred in the past) For the reviewer, the details of those locations are below:

-2 coordinates were located in areas that are close to healthcare, high risk (huanan_market, prince_wales_hospital).

-1 coordinate had average healthcare access, average risk (lvxi/Wang).

-4 coordinates had high risk and average healthcare access (Dafengkou/Wang, Tianjing/Wang, Guanxi/Li, yunnan/Li)

X	Y	Name	Obs
114.2620697	30.6166223	Huanan Seafood Market	SARS-Cov-2
114.1990333	22.3784089	Prince of Wales Hospital	SARS early cases
102.3197282	24.4800448	Dafengkou village	Wang locations in Yunnan
102.2732165	24.4871094	Lvxi village	Wang locations in Yunnan
97.8847234	24.8457588	Tianjing village	Wang locations in Yunnan
100.4010842	26.6312907	Guanxi	Li locations (HKU10-CoV)
100.2502033	25.5843648	Yunnan (no precision)	Li locations (SARSr-CoVs)

Overall I would recommend this paper for publication with minor revisions.
R. We appreciate your review, thank you.

REVIEWERS' COMMENTS

Reviewer #1 (Remarks to the Author):

I had greatly enjoyed reading the initial version of the manuscript, and I am fully convinced by the changes that the authors have made. I want to highlight the fact that the response to comments is particularly thorough, and the analyses that have been conducted have fully convinced me of the reliability of the results. As it stands, I have no reservation about this manuscript.

Reviewer #2 (Remarks to the Author):

I thank the authors for their revisions and clarifications, which I think have substantially improved the quality of the manuscript. Additional information on data and methods that are now included in the Supplementary Information have further addressed most of my concerns. I only have some minor comments.

I cannot find the revised caption (as stated by the authors in their response) for the Wilcoxon test.

On the discussion of methodological limitations, remote sensing data, in general, can be non-static so I find the sentence on static data sets and remote sensing in the discussion misleading. The accuracy of the Weiss et al. data set is known to be more precise in some areas than others. Can the authors please clarify whether this would vary for countries assessed in the study and how this may influence our interpretation of Figure 4?

Please check the grammar in the discussion.

Reviewer #3 (Remarks to the Author):

My comments/ suggestions have been responded to addressed thoroughly and thoughtfully, and I am happy to recommend the revised manuscript for publication.

RESPONSE TO REVIEWER COMMENTS

Reviewer #1 (Remarks to the Author):

I had greatly enjoyed reading the initial version of the manuscript, and I am fully convinced by the changes that the authors have made. I want to highlight the fact that the response to comments is particularly thorough, and the analyses that have been conducted have fully convinced me of the reliability of the results. As it stands, I have no reservation about this manuscript.

R: Thank you.

Reviewer #2 (Remarks to the Author):

I thank the authors for their revisions and clarifications, which I think have substantially improved the quality of the manuscript. Additional information on data and methods that are now included in the Supplementary Information have further addressed most of my concerns. I only have some minor comments.

R: Thank you.

I cannot find the revised caption (as stated by the authors in their response) for the Wilcoxon test.

R: Thank you. Here is the revised caption:

Fig. 4: Bivariate map showing the risk scores from hotspot data and access to healthcare. A. Black lines divide the limits for the 19 clusters identified by the multivariate spatial cluster analysis; Scenario 4 is represented in the map. B. Time to reach healthcare in areas where high emergent risk co-occurred far from healthcare for each of the four scenarios. Average values are in red, with p-values ~~for the differences shown~~ showing which pairwise comparisons differ from the null expectation of no difference in the Wilcoxon's test.

On the discussion of methodological limitations, remote sensing data, in general, can be non-static so I find the sentence on static data sets and remote sensing in the discussion misleading.

R: We acknowledge that the limitations apply to remotely sensed static data, but they could also apply to non-static data. Our discussion was more related to the limitations of the static data we used, as the environment is dynamic. We acknowledge that besides these limitations, our analysis provides a good picture of broad patterns with the best available data. We edited this sentence to read:

We acknowledge the complexity of pathogen responses to land use modification⁵⁵, and important limitations of our findings. The datasets used here are all static yet global and accessible. The static nature of the datasets is one limitation ~~of using remote sensing data in our assessment, as risk might vary temporally due to changes these datasets not apparent in static data~~. Hotspots may change in response to changes in economic and agricultural policies at national and subnational levels, international agreements such as Agenda 2030, and climate change adaptation⁵⁶. There are also several empirical data limitations. For instance, although the data sources for the health facilities being generally accurate³⁵, omission errors can occur ~~in static variables~~. Hospitals and clinics may close

The accuracy of the Weiss et al. data set is known to be more precise in some areas than others. Can the authors please clarify whether this would vary for countries assessed in the study and how this may influence our interpretation of Figure 4?

R: Thank you for pointing this out. As mentioned in Supplementary Table 1 in the previous version of this draft, we assume that there is good coverage in Asia, and according to the authors Google had the best data sources for Asian countries. China has the largest number of pixels with healthcare facilities in the world, followed by other countries considered in our analysis, such as India, Indonesia, Thailand, and Malaysia. Still, as our findings reveal, there is considerable variation in the access of healthcare, regardless of country.

After reading the reviewer's comment, we understand that we needed to clarify this information in the main text. Although we cannot provide a pixel-level calculation of omission errors for the entire travel time dataset, we now provide coverage data for all the countries in our study region and world totals.

We have edited the main text adding a note on accuracy and referencing Supplementary Tables 2 and 3 for extra details on coverage and accuracy. We believe that with the information presented in this reviewed version, the readers are now well equipped to interpret Figure 4.

Please find Supplementary Table 3 below:

Supplementary Table 3: Travel time and healthcare facility pixel count for the region of study based on the source data set (Weiss et al. 2020). A pixel is counted when it contains one healthcare facility or more.

Country	People per hospitals and clinics pixel	Hospitals and clinics pixel count	% Hospitals and clinics pixel count per world total
World total	19200	379231	100.000%
China	25900	53451	14.095%
India	52200	24136	6.364%
Indonesia	14800	17014	4.486%
Thailand	6900	9735	2.567%
Malaysia	10800	2769	0.730%
Philippines	43200	2358	0.622%
Vietnam	40100	2282	0.602%
Bangladesh	131400	1208	0.319%
Sri Lanka	25600	838	0.221%
Nepal	54300	582	0.153%
Myanmar	147500	339	0.089%
Singapore	23000	252	0.066%
Cambodia	80500	192	0.051%
Timor-East	9300	128	0.034%
Lao PDR	89400	76	0.020%
Bhutan	33100	24	0.006%
Brunei	36300	11	0.003%

Please check the grammar in the discussion.

R: Checked.

Reviewer #3 (Remarks to the Author):

My comments/ suggestions have been responded to addressed thoroughly and thoughtfully, and I am happy to recommend the revised manuscript for publication.

R: Thank you.